# Simple and Rapid Assembly of TALE Modules Based on the Degeneracy of the Codons and Trimer Repeats

**DOI:** 10.3390/genes12111761

**Published:** 2021-11-05

**Authors:** Lingyin Cheng, Xiaoqing Zhou, Yuling Zheng, Chengcheng Tang, Yu Liu, Shuwen Zheng, Yang Liu, Jizeng Zhou, Chuan Li, Min Chen, Liangxue Lai, Qingjian Zou

**Affiliations:** 1Guangdong Provincial Key Laboratory of Large Animal Models for Biomedicine, School of Biotechnology and Health Sciences, Wuyi University, Jiangmen 529020, China; chenglingyin163@163.com (L.C.); wyuchemzxq@126.com (X.Z.); zhengyulingwy@163.com (Y.Z.); wyuchemtcc@126.com (C.T.); 17865815973@163.com (Y.L.); Zheng_SW0@163.com (S.Z.); lichuan0718@126.com (C.L.); cmin0501@outlook.com (M.C.); 2Key Laboratory of Regenerative Biology, South China Institute for Stem Cell Biology and Regenerative Medicine, Guangzhou Institutes of Biomedicine and Health, Chinese Academy of Sciences, Guangzhou 510530, China; liu_yang@gibh.ac.cn; 3School of Biomedical and Pharmaceutical Sciences, Guangdong University of Technology, Guangzhou 510643, China; 13922169135@163.com

**Keywords:** transcription activator-like effectors, degenerated codon, trimer repeats, Gibson assembly

## Abstract

Transcription activator-like effectors (TALEs) have been effectively used for targeted genome editing, transcriptional regulation, epigenetic modification, and locus-specific DNA imaging. However, with the advent of the clustered regularly interspaced short palindromic repeat/Cas9 system, an easy-to-use tool with the same function as TALEs, TALEs have recently been abandoned because of their complexity, time consumption, and difficult handling in common labs. Here, we described a degenerated codon-based TALE assembly system for simple, rapid, and efficient TALE assembly. TALE trimers with nonrepetitive DNA sequences were amplified by PCR and sequentially assembled via Gibson assembly. Our method is cost-effective, requires only commonly used basic molecular biology reagents, and takes only 2 h from target sequence analysis to completion. This simple, rapid, and lab-friendly TALE assembly method will restore the value of TALEs in DNA targeting.

## 1. Introduction

The recent emergence of programmable DNA-binding proteins that are built on transcription activator-like effector protein (TALE) and clustered regularly interspaced short palindromic repeat (CRISPR)/Cas9 architectures has heralded a new era of efficient genome editing and manipulation. These tools provide new approaches to mastering and programming gene function and gene regulatory circuits and can be ultimately applied for the advancement of agriculture and medicine.

The DNA-binding domain of TALEs comprises an extended array of 34 amino acid (AA) repeats. Each repeat has one of four distinct repeat variable diresidues (RVDs) that differ only at AA positions 12 and 13 and confer nucleotide-binding specificity (NI to A, NG to T, HD to C, and NN or NH to G) [1,2]. TALEs can be fused with various effector domains, such as the endonuclease FokI, to generate TALE nucleases (TALENs), transcriptional regulatory domains (e.g., VP16 and KRAB) to form TALE-transcription factors (TALE-TFs) [3], or histone modifiers (e.g., LSD1) to construct TALE-chromatin editors (TALE-CEs) [4].

The CRISPR/Cas9 system is currently the most commonly used gene editing technology because it is inexpensive and easy to master and popularize [5,6]. However, it has notable off-target risks that make it unsuitable for complex genetic modification projects [7,8,9]. The guide RNA recognizes target DNA sequences that are only approximately 20 base pairs (bp) in length and thus easily forms mismatches, which lead to off-targets. Scientists have used point mutation technology to obtain SpCas9-HF1 [10], eSpCas9 [11], and HypaCas9 [12]. The accuracy of Cas9 gene editing has therefore been greatly improved. However, the potential off-target effects of Cas9 and its derivatives remain the biggest obstacle restricting the clinical application of the CRISPR/Cas9 system. Moreover, Cas9 technology cannot target any position in the genome due to the limitation of protospacer adjacent motif (PAM). Although the Cas9 mutants xCas9 [13], CAS9-NG [14], and CAS9-NRRH [15] have expanded the editable region of Cas9 in the genome, the limitation of PAM has not been eliminated.

Similar to CRISPR/Cas9, the TALEN system has efficient gene targeting capability. In addition, it has high accuracy and controllability. Its recognition range is a target sequence with a length of 10–31 (×2) bp and given that almost lacks off-target risks [16,17], it is closest to clinical application. The encoding sequence of TALEN is approximately 2000 bp to 3000 bp long and is thus far shorter than the encoding sequence of CAS9 and suitable for packaging in AAV viruses for further in-vivo treatments. Given that TALE effectors are proteins but not ribonucleoprotein (RNP) complexes, they can easily enter the mitochondria to edit the mitochondrial genome. Therefore, TALEs can be used in the clinical treatment of mitochondrial diseases [18,19]. Although the TALEN system has many advantages in gene editing and regulation, it is far less commonly applied than Cas9 technology, mainly because it is formed by multiple repetitive modules in series. Such a formation hinders most laboratories to complete the operation of TALEN technology by themselves.

A popular protocol for TALE construction is based on Golden Gate cloning, i.e., the stepwise cloning of TALEs by using type II restriction enzymes [20,21]. This procedure typically requires 3–5 days and depends on the time-consuming and cumbersome production of intermediate vector products. Alternative scalable methods based on solid-phase assembly, such as FLASH and ICA, have been developed [22,23]. These methods allow the rapid and automated production of TALEs but also require considerable expertise, equipment, and large plasmid libraries. Ligation-independent cloning [24], FairyTALE [25], and reTALE [26], provide alternative methods but ideally require automated liquid handling and have nontrivial initial setup and investment costs. STAR [27], which is based on Gibson assembly, is relatively rapid and scalable but also needs a plasmid library and has a complicated process and repeat length limitation.

A user-friendly and time-saving strategy for TALE assembly still cannot be satisfied, preventing most laboratories from applying TALE assembly in their own research. The ideal approach for TALE assembly would involve simple setup, high efficiency, minimal hands-on time, and high-throughput assembly without a plasmid library requirement. We devised degenerated codon-based TALE assembly (dcTA), which uses a novel strategy involving only several primer pairs, to address the above needs. The manual production of TALEs via PCR and assembly in a few hours can be achieved by using our method. Our method makes TALE assembly as convenient as CRISPR/Cas9 in plasmid construction. dcTA will enable labs to master TALE assembly, which can thus re-emerge as the main tool for genetic modification and regulation.

## 2. Materials and Methods

### 2.1. Plasmids

All TALE plasmid backbones were derived from pCAG-T7-TALENs with heterodimeric (ELD, KKR) domains obtained from Addgene (Plasmids #37184 and #40131). The CAG promoter was replaced by the EF1α promoter. A *LacZ* gene expression cassette with the Endonuclease Esp3I recognition site at both ends was inserted between the TALE N- and C-terminal domains. FokI, VPR, and EGFP were inserted after the C-terminus to obtain TALENs, TALE-transcription factors, and TALE-fluorescent proteins, respectively.

### 2.2. TALE Trimer Library Construction

As shown in Appendix A, we divided the primers used in TALE construction into two groups: primers for trimer library amplification and primers for adaptor trimers in order.

Four primers, which were named A2, T2, C2, and G2 and had four different sequences at their RVD sites, were used as templates, and T-F and T-R were used as primers to generate monomers. Four monomers were amplified via PCR. The four PCR products were further used as templates. The trimer library was amplified by using four forward primers—namely, A1, T1, C1, and G1, and four reverse primers—namely, A3, T3, C3, and G3. The trimer library contained a total of 64 (4  ×  4 ×  4) combinations.

### 2.3. Primer Library for Trimer Assembly in Order

Six primer pairs, 3  × 1, 3  × 2, 3  × 3, 3  × 4, 3  × 5, and 3  × 6, were located with a 21–23 bp overlap at the 5ʹ ends in order. The first primer 1F shared an overlapping sequence with the front end of the backbone vectors. The three last unique reverse primers—namely, R + 1, R + 2 and R + 3, were designed to target the first, second, and third monomers of the last trimer, respectively. The 5ʹ ends of these primers overlapped with the other ends of the backbone vectors.

### 2.4. TALE Assembly

The assembly process involved four main steps: target DNA sequence analysis, PCR, PCR amplicon purification, and isothermal assembly. Typically, the whole process takes only approximately 1–1.5 h to complete.

The first step is target DNA sequence analysis. In this step, the dcTA system could be used to target arbitrary dsDNA with lengths of 10–21 bp. The target nucleotides (the number is marked n) were divided into several triplets in order. The last group may have only one or two nucleotides. A total of x groups (x = [(n + 2)/3]) were present.

The second step is PCR, wherein in accordance with each triplet nucleotide, the corresponding template was sequentially taken from the TT library. Then, 3M1 to 3M (x − 1) primer pairs were needed to amplify the first x − 1 trimer modules, and Zn primer pairs were used to amplify the last module. PCR was performed by amplifying each TT by using the corresponding primers in individual tubes.

The third step is PCR amplicon purification. After PCR reaction, all PCR amplicons were mixed together and purified by using a Monarch^®^ PCR & DNA Cleanup Kit (NEB, Beijing, China) within 10 min.

The last step is TALE module assembly. TALE backbone vectors with TALE N- and C-terminals were cut by BbsI in the middle. The vectors and the PCR amplicons were mixed at a volume of 0.02 pmol each and added with isothermal recombinase. The samples were incubated in a thermocycler at 37 °C (ClonExpress MultiS One Step Cloning Kit, Vazyme, Nanjing, China) or 50 °C (Gibson Assembly^®^ Master Mix, NEB, Beijing, China) in accordance with the types of recombinases for 15–20 min. Fully assembled plasmids were generated in the reaction and then directly transformed into 25 μL of chemically competent bacteria and plated for overnight growth.

### 2.5. Blue–White Screening

The TALE backbone vectors were cut by Esp3I to remove the *LacZ* cassette and recombined with trimers. Luria–Bertani (LB) solid agar media containing ampicillin sodium (100 μg/mL), IPTG (1 mM), and X-Gal (200 μg/mL) were prepared before bacterial transformation. The recombinant plasmids were spread on the transformed competent cells and incubated inverted at 37 °C for 16 h. The white colonies were picked for further analysis.

### 2.6. Assembled Plasmid Validation

Bacterial colonies were picked up and cultured for another 8 h in liquid LB medium. Plasmids were extracted by using QIAprep Spin Miniprep Kit (Qiagen, Dusseldorf, Germany). The assembled plasmids were verified by using PstI/BamHI or SphI/SnaBI restriction digestion. Full-length products were sequence-verified through Sanger sequencing.

### 2.7. Cell Culture and Transfection

HEK293T and U2OS cell lines were grown in Dulbecco’s Modified Eagle’s medium supplemented with 10% fetal bovine serum. HEK293T and U2OS cells were transfected by using Lipofectamine™ 3000 Transfection Reagent (Thermo, Shanghai, China). TALENs were used to target the *AAVS1* site of the HEK293T cells, and TALE-EGFP was applied to target the telomerases of the U2OS cells. TALE-VPR was targeted the promoter of the endogenous vascular endothelial growth factor (*VEGF*) gene of the HEK293T cells. The experiment was conducted in accordance with the manufacturer’s instructions.

### 2.8. ELISA

Two TALE-VPRs, which were named VPR1 and VPR2, were designed to target the ~500 bp downstream of *VEGF-A* TSS. The HEK293T cells were transfected. Cell media were harvested 48 h after transfection, and secreted VEGF-A protein levels in the media were assayed by using a human VEGF-A ELISA kit (R&D Systems, Shanghai, China). All samples were measured in accordance with the manufacturer’s instructions.

### 2.9. Rabbit Embryo Injection

The details of the procedures for the microinjection of pronuclear-stage embryos are provided in previously published protocols [28]. In brief, both TALENs (100 ng/µL) were coinjected into the cytoplasm of pronuclear-stage zygotes in the experimental group. The embryos were then transferred into Earle’s balanced salt solution medium and cultured at 38.5 ℃ in 5% CO_2_.

### 2.10. DNA Extraction and Targeted Amplicon Sequencing

The injected rabbit embryos were collected at the blastocyst stage and lysed in 10 μL of lysis buffer (0.45% NP-40 plus 0.6% proteinase K) at 56 °C for 60 min and then at 95 °C for 10 min and subsequently subjected to Sanger sequencing. Flow-sorted cells were identified by using the same method. The target sequence was amplified from the genome of the transfected cells by PCR (2 × PhantaMax Master mix, Vazyme, Nanjing, China) with specific primers. The products were then sent out for Sanger sequencing and deep sequencing.

### 2.11. Targeted Deep Sequencing

Targeted sites were amplified from the extracted DNA with the corresponding site-specific primers by using Q5 High-Fidelity DNA Polymerase (Takara, Beijing, China). In this experiment, we used two deep-sequencing platforms for on-target analysis. The identification primer (18–20 nt) was designed in accordance with the principles of conventional PCR primer design. Specifically, the detection site must be within the range of 10–100 bp in the forward or reverse primer, and the bridging sequence 5′–ggagtgagtacggtgtgc–3′ should be added to the front of the forward primer. At the same time, the bridging sequence 5′–gagttggatgctggatgg–3′ was added to the front end of the reverse primer. Each PCR was performed with a 30 μL volume comprising 2 μL of the template in accordance with the manufacturer’s protocol. The paired-end sequencing of PCR amplicons was performed by Illumina MiSeq.

## 3. Results

### 3.1. Construction of TALE Trimers

We know that the TALE repeat contains 34 AAs with RVD at positions 12 and 13 (NI, NG, NN, HD, Figure 1A). Accordingly, the DNA template for the TALE repeat contained 102 nucleotides. We first constructed 64 TALE trimers by PCR to construct an arbitrary TALE sequence. The trimers contained 34 repeat AAs with RVD at positions 12 and 13. We set the gene of trimers wherein nucleotide variations replaced synonymous codons (Appendix AA). The degenerated codon could reduce the number of repetitive sequences and lower the rock for TALE construction. We performed two rounds of PCR to construct these 64 templates (Figure 1B). In the first round of PCR, four primers, which were named A2, T2, C2, and G2 and had four different sequences at their RVD sites, were used as the templates, and T-F and T-R were used as the primers. The four PCR products were further used as the templates for the second round of PCR. The trimer library was amplified by using the four forward primers A1, T1, C1, and G1 and the four reverse primers A3, T3, C3, and G3. The trimer library contained a total of 64 (4  ×  4 ×  4) combinations (Appendix AB).

### 3.2. Rapid Assembly of TALE Repeats with the dcTA System

We exploited codon degeneracy to redesign the coding sequences at both ends of the trimer modules. Therefore, we generated six ordered primer pairs (OPPs)—namely, 3  × 1, 3  × 2, 3  × 3, 3  × 4, 3  × 5, and 3  × 6, each with a 21–23 bp overlap at their 5ʹ ends in order (Figure 1C and Appendix AC). The first primer 1F shared an overlap sequence with the front end of the backbone TALE vectors. This sequence was cut by Esp3I. The three last unique reverse primers R + 1, R + 2, and R + 3 were designed to target the first, second, and third monomers of the last trimer-module, respectively. The 5′ ends of these primers overlapped the other end of the backbone vectors. As described in Appendix AC, the last three reverse primers paired with the forward primers 4F, 5F, 6F, and 7F for a total of 12 combinations, which were designated as Z10–Z21. The trimers from the library were used for PCR with the OPPs, and the products were constructed into linearized backbone vectors via isothermal assembly (Figure 1D).

The assembly process comprised four main steps: target DNA sequence analysis, PCR, PCR amplicon purification, and isothermal assembly. Typically, the whole process took only approximately 1–2 h to complete (Figure 2A). The construction of the TALE vector for targeting TTTCTGTGACCAATCCT was used as an example. First, the target sequence was sequentially divided into TTT, CTG, TGA, CCA, ATC, and CT. Second, the corresponding trimers TTT, CTG, TGA, CCA, ATC, and CTN (N represents any base) were removed from the library and used as templates in order. PCR amplification was performed sequentially with the OPPs (Figure 2A). The TALE backbone plasmid was digested with Esp3I to remove the *LacZ* gene. Third, the linearized vectors and PCR fragments were purified and assembled via Gibson recombination to form TALE expression vectors. Lastly, bacterial transformation and blue–white selection were performed to generate correct and abundant TALE vectors. On the second day, more than 99% of bacterial colonies were white, and only a few colonies were blue (Figure 2B). Fifteen white clones were selected for further expansion. The plasmids were extracted and digested by enzymes. Fourteen out of 15 (93.3%) exhibited the expected bands on gel electrophoresis (Figure 2C).

We constructed a TALE module targeting the *EGFP* gene with lengths of 12 bp to 21 bp (Figure 2D) to verify the capability of the dcTA approach to assemble TALE repeats with different lengths. Enzyme digestion showed that all 10 repeats with different lengths could be constructed into TALE backbones (Figure 2E). This result suggested that the simple, rapid, and flexible assembly of TALE repeats will be very useful for various applications.

### 3.3. TALEs Generated by dcTA System Are Functional

We next tested whether the dcTA TALE could target the respondent DNA in mammalian cells. First, the reporter vector containing Puro–Target–sfGFP was constructed. After transfection into eukaryotic cells, the expressed *Puro* resistance gene was fused with a few nonfunctional AAs coded by the target, which had the terminal code TAA at the end of the target region. The *sfGFP* gene followed the target and was out of frame with the previous *Puro* resistant gene. Next, a pair of TALENs with the FokI endonuclease domain were designed in accordance with the target site in the reporter gene. spCas9 with gRNA targeting the target site was constructed as the control. In theory, both engineered endonucleases break the DNA at the target site. This phenomenon is then followed by nonhomologous DNA end joining, which leads to insertions or deletions (indels) and ultimately results in frameshift mutation and sfGFP expression (Figure 3A). The reporter and engineered endonucleases were transfected into HEK293T cells to prove that dcTA TALEN was functional. TALEN and spCas9 caused approximately 25% of the total cells to express green fluorescence (Figure 3B). dcTA TALEN RNAs were further synthesized via in-vitro transcription to target the endogenous rabbit *DMD* gene. The paired TALENs were injected into rabbit embryos (Figure 3C). Blastocysts were collected for PCR and sequencing 4 days after injection. Sanger sequencing detected mutations in four out of six of the embryos. Targeted deep sequencing further showed that 28.3% indels from −17 bp to +13 bp existed in all four embryos (Figure 3D). This result indicated that dcTA TALEN gene editing has functions in cells and embryos.

Second, to use the dcTA TALE in endogenous genomic loci in living cells, the TALE-sfGFP vector was designed to target the human telomere repeat TTAGGG (Figure 3E). The vector was transfected into U2OS cells. Dozens of green dots were observed in the nuclei of the cells. By contrast, the normal EGFP was distributed in all cell bodies (Figure 3F). This result indicated that dcTA TALE-sfGFP bound telomeres efficiently in living cells.

Finally, to test whether dcTA TALE could be used to activate endogenous genes, we designed TALE for the activation of the human *VEGF* gene (Figure 3G). The *VPR* fusion gene was inserted downstream of TALE, which was designed to target the promoters of the endogenous human *VEGF-A* gene. The TALE-VPRs TV1 and TV2 were designed and transfected into HEK293T cells. The VEGF factors in the supernatant were detected by ELISA 48 h later. VEGF released by TV1- and TV2-transfected cells was nearly double that of the wild-type HEK293T cells (52.94 and 56.04 ng/mL, respectively). VEGF released by TV1- and TV2-co-transfected cells was 5 times (160.39 ng/mL, Figure 3H) that of the wild type. These results indicated that dcTA TALE can be used for endogenous gene regulation.

## 4. Discussion

Here, we reported a user-friendly, simple, cheap, and rapid approach for TALE assembly. Our approach exploited degenerated codons to generate nonrepetitive trimers. TALE repeats were reduced three-fold by using trimers and can be constructed with high efficiency via isothermal assembly. They can be constructed with arbitrary lengths and recognize substrates specifically. In this research, TALE was used to target indels, image telomeres in living cells, and activate the endogenous *VEGF* gene. Remarkably, only approximately 30% of the TALEs had point mutations.

This problem will be solved by using new primers with high purity and PCR enzymes with high fidelity. The routine production of TALE will complement emerging reagents and open up new possibilities for a myriad applications wherein gene regulatory circuits need to be interrogated or engineered.

TALE, an important complement of the CRISPR/Cas9 system, has been applied widely in gene editing, gene regulation, and chromatin visualization. It has incomparable advantages, specially in mitochondrial DNA and heterochromatin DNA editing [29], and is sufficiently small for the construction of the AAV virus for therapeutic applications. In this work, we established dcTA, a novel method that is very simple and easy to master in general laboratories.

## Figures and Tables

**Figure 1 genes-12-01761-f001:**
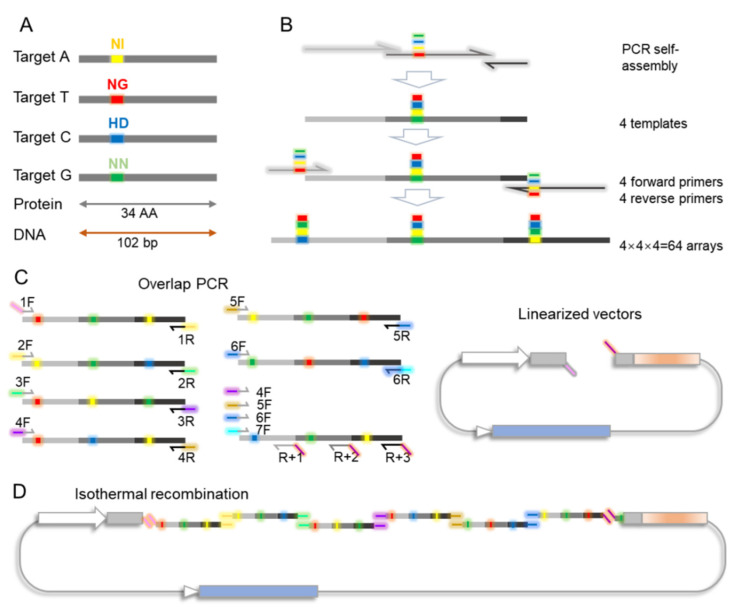
Schematic of TALE construction. (**A**). One TALE repeat domain contained 34 AAs with one of the four different RVDs. Correspondingly, the coding DNA sequence contained 102 bp. The colored sites represent four different RVDs. (**B**). PCR strategy for the amplification of 64 arrays of trimers. Half arrows represent primers, and colored sites represent the four different RVD coding sites. Three thick lines with different shades of gray represent TALE monomer repeat coding sequences with synonymous codons. (**C**). PCR strategy for the amplification of trimers with 21 bp overlapping ends in order. Each color on the primer represents a unique sequence with synonymous codons. Vectors can be linearized by restriction endonucleases. (**D**). Strategy for TALE vector assembly by using isothermal recombination. The trimers with ordered ends are recombined in order into linearized vectors.

**Figure 2 genes-12-01761-f002:**
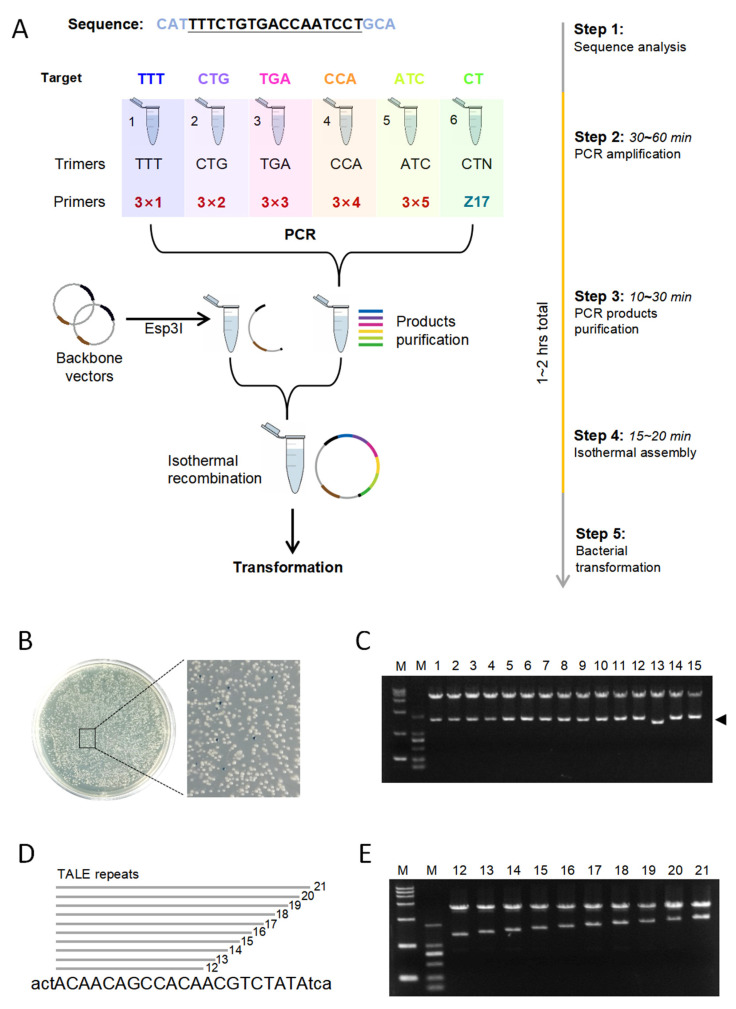
Construction of TALE vectors with different repeats. (**A**). Summary of the 2 h assembly of the TALE vector with the dcTA protocol by using PCR and isothermal recombination. (**B**). Blue–white screening for TALE vector construction. White bacterial colonies accounted for more than 99% of the colonies. (**C**). Clones were characterized by restriction digestion, which yielded a 1.8 kb fragment (arrowhead) for fully assembled TALE vectors. M, 15,000 bp and 2000 bp marker; 1–15, the identified colonies. (**D**). Ten TALE vectors were designed to target *EGFP* gene. The vectors have different lengths of 12–21 repeats. (**E**). TALE vectors with different lengths were cut through restriction digestion and yielded restrictions with lengths of 1400–2300 bp. M, 15,000 bp and 2000 bp marker; 12–21, the identified vectors with 12–21 TALE repeats.

**Figure 3 genes-12-01761-f003:**
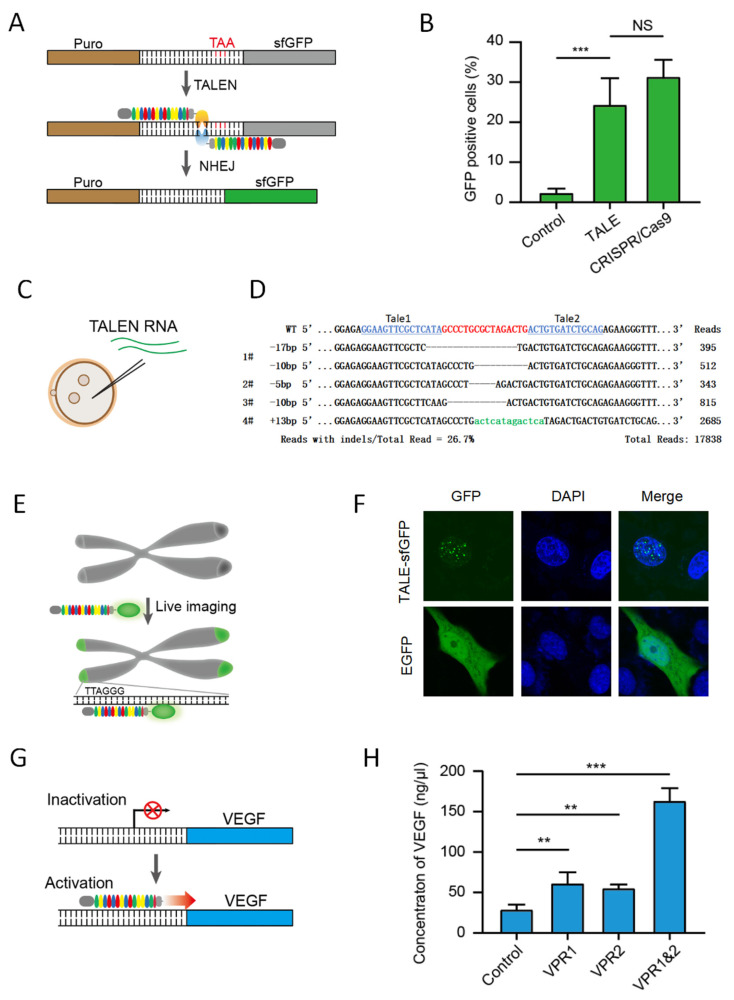
TALEs generated by the dcTA system are functional. (**A**) Schematic of the TALEN editing of the reporter gene. Brown, puromycin resistant gene (*Puro*); TAA, stop codon; Gray, silenced *sfGFP* gene; Green, *sfGFP* in frame with *Puro*. Fluorescence ratio of report in HEK293T cells. (**B**) Quantification of EGFP activation in HEK293T cells transfected with TALENs generated by dcTA. TALEN and CRISPR/Cas9 targeted the upstream of the stop codon between *Puro* and *sfGFP*. (**C**) Schematic of TALEN RNA injection into rabbit embryos. (**D**) Amplicon sequencing data showing the most frequent indels in the targeted region within the *DMD* gene locus. The WT sequence is shown at the top with the target sites highlighted in red and TALE pair-binding sites highlighted in blue. The sequence highlighted in green are the insertion bases. (**E**) Schematic of the live imaging of telomeres with TALE-sfGFP. (**F**) Confocal imaging of U2OS cells transfected with either EGFP or TALE-sfGFP. (**G**) Mechanism of the TALE-VPR targets at the upstream of the TSS of endogenous *VEGF-A* gene to activate gene expression. (**H**) Quantification of VEGF concentration in the cell culture supernate. Asterisks indicate a significant difference between two group (** *p* < 0.01, *** *p* < 0.001).

## Data Availability

All data used in this paper are available in the article and supplementary materials.

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
