# Peer review of "Simple and Rapid Assembly of TALE Modules Based on the Degeneracy of the Codons and Trimer Repeats"

_genes, 2021, doi:10.3390/genes12111761_

Round 1

Reviewer 1 Report

The manuscript by Cheng et al. entitled “Simple and rapid step for TALE assemble based on the degenerate codon of trimer repeats” describes a simplified method to synthesise TALENs.

There are many published methods to synthesise TALENs, yet the hands-on time and complexity of the associated molecular cloning make them difficult to apply in a small laboratory setting.

The authors capitalised on previous methods to establish a rapid assembly that takes advantage of:

  • The degeneracy of the codon code.
  • Isothermal assembly (in lieu of PCR).

Overall, the technique seems pretty straightforward and amenable to most labs, although the overall merit is limited since it is now easy to commercially purchase highly active and sequence verified TALEN mRNA.

The authors achieved genome editing, epigenome activation and genome visualisation using their technique.

However, several limitations must be addressed to improve the manuscript.

  • The command of the English language is limited, which makes the reading difficult at times. We advise a native speaker to review and edit the entire manuscript. Likewise, the title itself contains many typos. We advise the authors to change it to “Simple and rapid assembly of TALE modules based on the degeneracy of the codons and trimer repeats”.

All along the manuscript, “degenerate” should be replaced with “degenerated” (this occurs several times).

  • The method developed herein capitalises on a similar method that uses Golden Gate assembly for TALEN production: A Transcription Activator-Like Effector (TALE) Toolbox for Genome Engineering (Sanjana et al. Nat Prot. 2012). This should be referenced in the manuscript.
  • The last step of the procedure (i.e. TALE assembly) is performed using isothermal recombinases. Although advantageous, the authors mentioned that the fragments were assembled “in accordance with the types of recombinases” (line 139). This information is not sufficient to inform the readers who intend to use this method. Examples of the isothermal recombinases used (e.g. NASBA, LAMP, RCA, RPA…) should be given and the rationale for using each type of recombinase needs to be explained.
  • In 2.11 the authors mentioned deep sequencing for off-target analysis, yet these results are not provided anywhere. What were the criteria for selecting potential off-target candidates and what is the outcome of the off-target analysis?

Reviewer 2 Report

The authors proposed the dcTA technique, an effective TALE assembly method. The authors made 64 modules that recognize trimers of all combinations using PCR and produced TALEs very easily using isothermal recombination. By applying TALE to gene editing, gene regulation, and site-specific imaging, it has been shown that effective TALE can be produced through dcTA technology. The experiment was performed properly, and the content of the paper was good.

Minor concerns

In Figure 2C, 2E.; Add marker size information to Figure

In Figure 2E:  I suggest using a low % agarose gel to further show differences between TALE repeats (12-21).
